# NUPR1: A Critical Regulator of the Antioxidant System

**DOI:** 10.3390/cancers13153670

**Published:** 2021-07-22

**Authors:** Can Huang, Patricia Santofimia-Castaño, Juan Iovanna

**Affiliations:** Centre de Recherche en Cancérologie de Marseille (CRCM), INSERM U1068, CNRS UMR 7258, Aix-Marseille Université and Institut Paoli-Calmettes, Parc Scientifique et Technologique de Luminy, 163 Avenue de Luminy, 13288 Marseille, France; can.huang@inserm.fr (C.H.); patricia.santofimia@inserm.fr (P.S.-C.)

**Keywords:** NUPR1, cell death, ferroptosis, ROS, cell stress

## Abstract

**Simple Summary:**

Nuclear protein 1 (NUPR1) is activated in cellular stress and is expressed at high levels in cancer cells. Much evidence has been gathered supporting its critical role in regulating the antioxidant system. Our review aims to summarize the literature data on the impact of NUPR1 on the oxidative stress response via such a regulatory role and how its inhibition induces reactive oxygen species-mediated cell death, such as ferroptosis.

**Abstract:**

Nuclear protein 1 (NUPR1) is a small intrinsically disordered protein (IDP) activated in response to various types of cellular stress, including endoplasmic reticulum (ER) stress and oxidative stress. Reactive oxygen species (ROS) are mainly produced during mitochondrial oxidative metabolism, and directly impact redox homeostasis and oxidative stress. Ferroptosis is a ROS-dependent programmed cell death driven by an iron-mediated redox reaction. Substantial evidence supports a maintenance role of the stress-inducible protein NUPR1 on cancer cell metabolism that confers chemotherapeutic resistance by upregulating mitochondrial function-associated genes and various antioxidant genes in cancer cells. NUPR1, identified as an antagonist of ferroptosis, plays an important role in redox reactions. This review summarizes the current knowledge on the mechanism behind the observed impact of NUPR1 on mitochondrial function, energy metabolism, iron metabolism, and the antioxidant system. The therapeutic potential of genetic or pharmacological inhibition of NUPR1 in cancer is also discussed. Understanding the role of NUPR1 in the antioxidant system and the mechanisms behind its regulation of ferroptosis may promote the development of more efficacious strategies for cancer therapy.

## 1. Introduction

The human NUPR1 gene, located on chromosome 16, codes for a low molecular weight protein (8 kDa), NUPR1, which lacks a stable secondary and tertiary structure, and is expressed at low levels in healthy cells under normal physiological conditions [1]. However, NUPR1 can be transcriptionally activated under stress conditions such as oxidative stress, ER stress, and metabolic stress, to protect the cells from these stresses [2]. Compared with normal cells, cancer cells, particularly those in the tumor microenvironment, show higher basal levels of cellular stress such as oxidative stress and ER stress [3,4]. Thus, NUPR1 is present at correspondingly higher levels in cancer cells [5]. Oxidative stress refers to elevated levels of ROS that damage biomacromolecules such as proteins, DNA, and lipids in cells [6]. In order to protect themselves from such ROS-mediated tumor cell damage or cell death, cancer cells up-regulate the antioxidant system [7]. In cancer cells, ER stress may activate the ubiquitin–proteasome system (UPS), the unfolded protein response (UPR), or other cytoprotective mechanisms to restore homeostasis and increase the adaptability to the adjacent environment [8,9,10]. Methamphetamine (METH)-induced ER stress has been shown in non-cancer cells to initiate apoptosis and autophagy via the NUPR1/CHOP/TRIB3 pathway [11,12,13]. Interestingly, ER stress can be triggered by high levels of ROS, which emphasizes the role of NUPR1 in the crosstalk between different stress responses [14]. Moreover, increased aerobic glycolysis and oxidative stress are important features of metabolism in cancer cells [15]. NUPR1 is a highly expressed protein that confers drug resistance to cancer cells through the maintenance of redox and the antioxidant system in various pathological conditions [16,17]. Increasing evidence shows that NUPR1 is activated under high ROS levels to protect cells against oxidative damage. Recently, ferroptosis, an oxidative cell death mechanism, was discovered in V-Ki-ras2 Kirsten rat sarcoma viral oncogene homolog (KRAS) mutant tumor cells, and is characterized by antioxidant defense system failure and the induction of lipid peroxidation [18]. Ferroptosis is the result of a redox imbalance caused by the cancer cells’ incapacity to maintain cell metabolism and cellular functions though antioxidant defense [19]. It does not occur frequently due to the strong antioxidant defense mechanism present in cancer cells [20]. Previously, the GSH/GPX4 system was thought to be the only regulator of ferroptosis [21,22]. Later, more signals were found to suppress ferroptosis, such as COQ10/FSP1 [23,24] and BH4/DHFR [25]. The mounting evidence supporting ferroptosis as a therapeutic target has led to the development or identification of more ferroptosis inducers for cancer treatment [26]. They have proven useful in overcoming drug resistance and preventing tumor metastasis formation [27], and may represent a promising strategy in combination with other anti-cancer therapies. One important process in cancer cells allowing the formation of drug resistance and heterogeneity is the upregulation of NUPR1 expression [28]. The NUPR1 was recently shown to participate towards iron and energy metabolism and protect cells or tissues against ferroptosis [29,30]. Nevertheless, how NUPR1 is transcriptionally activated in the oxidative stress response and its exact role in ferroptosis remain unclear. In this review, we aimed to clarify the role of NUPR1 in the antioxidant system and redox balance. We provide arguments favoring the targeting of NUPR1 to induce ferroptosis as a therapeutic strategy for treating cancer.

## 2. Oxidative Stress Activates NUPR1

NUPR1 expression is associated with the deregulation of ROS levels in cells, and has been shown to be promoted with elevated ROS levels. For example, by increasing ROS levels, quercetin can induce the expression of NUPR1 and activates autophagy in osteosarcoma cells [31]. Either supplementation of the antioxidant N-acetyl cysteine (NAC) or knockout of NUPR1 can inhibit the autophagy and lipid peroxidation induced by quercetin, thus implicating the ROS/NUPR1 pathway in the mechanism used by quercetin [31]. Similarly, the cytotoxic heavy metal cadmium (Cd) can decrease glutathione (GSH) levels, causing oxidative stress and lipid peroxidation in cancer cells [32,33]. In an oral squamous carcinoma xenograft model, the Cd led to lipid peroxidation and induced NUPR1-dependent autophagy, which was again inhibited by NAC [34]. However, how ROS regulates the expression of NUPR1 in this context remains an open question.

### 2.1. Oxidative Stress Regulates NUPR1 Expression via a ER Stress-Mediated Pathway

NUPR1 can be transcriptionally activated under oxidative stress or ER stress conditions. ROS as well as both oxidants and reducing agents have been shown to disrupt protein folding and induce calcium release from the ER, leading to ER stress [35,36]. For example, intracellular ROS is sufficient to trigger the calcium ions (Ca^2+^) release from the ER, inactivate Ca^2+^-dependent ER partners (calnexin and calreticulin), and induce the ER stress response [14,37,38]. Concordantly, NUPR1 is activated in the ER stress response caused by intracellular high ROS levels. For example, the neurotoxic drug methamphetamine (METH) causes mitochondrial dysfunction, increases oxidative stress, changes intracellular Ca^2+^ dynamics, and thereby activates the ER stress response [39]. Xu and colleagues showed that apoptosis and autophagy induced by METH in pheochromocytoma are the consequence of the upregulation of NUPR1 during ER stress [11]. METH also activates the downstream pathways (CHOP/Trib3) of NUPR1 in ER stress, inhibits the mechanistic target of rapamycin (mTOR) phosphorylation, and thereby promotes neuronal autophagy [12]. Thus, Nupr1/CHOP/Trib3 signaling appears as a potential therapeutic target for drug-induced neurotoxicity. The involvement of Ca^2+^ is supported by evidence of the calcium chelator BAPTA-AM significantly inhibiting elevated NUPR1 mRNA levels in liver cancer cells treated with hydrogen peroxide (H_2_O_2_) [40]. Altogether, these results imply that the transcriptional activation of NUPR1 regulated by ROS is related to the ER stress triggered by Ca^2+^ (Figure 1).

### 2.2. Oxidative Stress Mediates NUPR1 Expression through Non-ER Pathways

Several studies have shown that the elevated NUPR1 level in oxidative stress is likely to be directly regulated by some factors in the non-ER stress pathway. For example, a ferroptosis inducer tert-butyl hydroperoxide (TBHP) can activate NUPR1 through the oxidative stress pathway, but not the ER stress pathway [42,43]. High intracellular ROS levels caused by H_2_O_2_ activate the expression of activating transcription factor 3 (ATF3), which then binds to the NUPR1 promoter to activate transcription [41]. Another study showed that NUPR1 was not activated in liver cancer cells exposed to H_2_O_2_ for a short time, but that NUPR1 protein levels increased over time accompanied by mitochondrial defects [40]. NUPR1 levels seem therefore to positively correlate with the ROS levels. In support of this notion, ROS and the NUPR1 were decreased in HERV-K env knockout colorectal cancer, compared with the wild-type cells; however, HERV-K re-expression restored the elevated levels of both NUPR1 and ROS [49]. Although ER stress can be initiated by ROS accumulation, ROS or Ca^2+^ release is insufficient to induce a severe ER stress response [50]. We recently found that the eukaryotic initiation factor 2α (eIF2α) phosphorylation and its downstream signaling pathways in ER stress require the participation of NUPR1 [51]. Indeed, our previous studies have shown that many proteins that participate in ER stress show minimal activation in the absence of NUPR1 [52]. While inhibition of NUPR1 significantly increases the ROS production, it dramatically inhibits the ER stress response, indicating that NUPR1 is a key mediator of the crosstalk between oxidative stress and ER stress (Figure 1).

## 3. NUPR1 Controls Redox Homeostasis and Protects Mitochondria

NUPR1 is a functional cellular stress protein which plays an important role in controlling its downstream signaling pathways and redox reactions. For example, oxidative damage to mitochondrial DNA and mitochondrial dysfunction caused by exposure of cells to the ferromagnetic metal nickel (Ni) [53], is associated with activated NUPR1 transcription through activator protein 1 (AP-1) binding to the NUPR1 promoter, in human bronchial epithelial cells. The NUPR1 thereby promotes the transformation of healthy cells into cancer cells as a protective mechanism against oxidative stress [45]. Knockdown of NUPR1 significantly reduced the cell viability and clonogenic ability of human bronchial epithelial cells exposed to Ni [45]. Therefore, high NUPR1 levels protect against oxidative damage, especially in those cells such as cancer cells with high ROS levels.

### 3.1. NUPR1 Regulates the Antioxidant System

The nuclear factor erythroid 2-related factor 2 (Nrf2) is an important transcription factor that regulates cellular oxidative stress response, and is also a central regulator that maintains intracellular redox homeostasis [54]. Nrf2 can reduce oxidative damage by regulating the constitutive expression of several antioxidant genes, and maintaining cellular redox homeostasis [55]. The NUPR1-mediated antioxidant pathway is, however, independent of NRF2 in the antioxidant system. Indeed, knockdown of NUPR1 triggers the expression of lipid detoxification genes such as aldo-keto reductase family 1 member C1 (AKR1C1) in keratinocytes and pancreatic cancer cells but does not affect NRF2 expression or nuclear translocation [56]. Inversely, knockdown of NRF2 showed no effect on the expression of either NUPR1 or AKR1C [56]. In another study, NUPR1 transcriptionally activated the presynaptic ROS sensor synaptosome associated protein 25 (SNAP25) and maintained the autolysosomal efflux in breast cancer cells [57,58]. In short, these studies emphasize the unique role of NUPR1 in the antioxidant system.

### 3.2. NUPR1 Maintains Mitochondrial Function

As the center of energy and the source of most ROS, mitochondria play an indispensable role in redox reactions [59]. The latest research shows that NUPR1 is involved in the regulation of mitochondrial function [40]. NUPR1 expression is usually induced in cells suffering persistent oxidative damage and mitochondrial dysfunction. For example, fascaplysin, a selective cyclin-dependent kinase 4 inhibitor with antitumor activity, can directly trigger mitochondrial depolarization and ATP consumption [60]. Studies showed that cell death induced by fascaplysin is accompanied by mitochondrial dysfunction, which causes the decreased mitochondrial membrane potential (MMP) and high ROS levels, and promotes NUPR1 expression as a protective defense [61]. However, knockdown of NUPR1 reduced the mitophagy induced by fascaplysin, and promoted greater ROS production [61]. These results indicate that the overexpression of NUPR1 is a protective process when mitophagy occurs in cells. In fact, NUPR1 inactivation causes mitochondrial dysfunction, antioxidant system failure, and accelerates ROS accumulation. Our previous study also showed that the inactivation of NUPR1 induced mitochondrial dysfunction including Ca^2+^ outflow, decreased ATP levels, and ROS increase in pancreatic cancer cells [52,62]. At the same time, the inactivation of NUPR1 reduced the expression of the tricarboxylic acid cycle (TCA) genes and activated glycolysis genes, thereby promoting a strong metabolic reprogramming in cancer cells [52].

### 3.3. NUPR1 Regulates Energy Metabolism

Mitochondria are the hub of energy metabolism, mainly using the energy from glucose, glutamine and fatty acid [63]. NUPR1 regulates cell energy metabolism by maintaining or enhancing the mitochondria function through different signal pathways. For instance, γ-H2AX, a histone that regulates ROS-mediated DNA damage through the Nox1/Rac1 pathway, was highly increased in NUPR1 knockdown cells subjected to hypoxia or glucose starvation [46,47,64]. The re-expression of NUPR1 or supplementation with NAC prevented oxidative DNA damage and reduced the γ-H2AX levels. Inhibition of aurora kinase A (AURKA), a gene associated with DNA damage in autophagy and that regulates mitochondrial dynamics and energy production, causes oxidative stress and subsequent ferroptosis in gastrointestinal cancer cells [65,66]. Intriguingly, hypoxia or glucose starvation can cause a decrease in AURKA levels with an increase in NUPR1, while knockdown of NUPR1 further significantly reduces the transcription and expression levels of AURKA, indicating that the overexpression of NUPR1 offsets the transcription changes during metabolic stress [46]. Glucose starvation also activates the molecular chaperones glucose-regulated protein 75 (GRP75) and glucose-regulated protein 94 (GRP94) because of a glycosylation defect-induced ER stress [67]. Elsewhere, NUPR1 has been shown to act as a co-activator of peroxisome proliferator-activated receptor-gamma coactivator 1-alpha (PGC-1α), a master regulator of ROS scavenging enzymes and an effective stimulator of mitochondrial biogenesis and respiration [68]. PGC-1α is responsible for the transcription of mitochondrial transcription factor A (TFAM), thereby maintaining the replication and transcription of mitochondrial genes [69]. The overexpression of PGC-1α leads to changes in the expression of the PGC-1α responsive gene fatty acid synthase (FAS) in prostate cancer [70]. Altogether, these studies demonstrate that NUPR1 regulates a series of signaling molecules to maintain mitochondrial function and energy metabolism.

In vivo studies have also underlined the important role of NUPR1 in energy metabolism [71,72]. For example, while wild-type mice showed impaired glucose tolerance under a short-term high-fat diet (HFD), NUPR1 knockout mice maintained normal glucose tolerance even after 16 weeks of HFD [48]. Elsewhere, compared with wild-type mice, NUPR1 knockout mice have increased β cell mass and higher insulin secretion by pancreatic islets [73]. Another study showed that following the HFD diet, NUPR1 knockout mice showed a significantly increased mass of islets, total number of islets, and average islet size, each indicative of protective measures against obesity, glucose intolerance, and insulin resistance [71]. These studies seem to indicate that NUPR1 knockout mice have stronger glucose tolerance and stronger glucose metabolism, suggesting that NUPR1 regulates energy metabolism in vivo.

Together, these data highlight the importance of high NUPR1 levels in maintaining energy metabolism and the antioxidant system. Compared with healthy cells, cancer cells have higher ROS levels due to their uncontrolled metabolic capacity during hyperproliferation [74,75]. The inactivation of NUPR1 in cancer cells can trigger ROS overproduction by inducing mitochondrial dysfunction, thereby leading to cell death [52]. One such potent inhibitor of NUPR1 is ZZW-115, which causes mitochondrial damage, energy metabolism deregulation, and ROS overproduction in a variety of cancer cells [62,76]. Altogether, NUPR1 exerts a protective role against oxidative damage by maintaining mitochondrial function and redox reactions in cells with high ROS levels.

## 4. NUPR1 Is a Key Factor of Ferroptosis

Increasing evidence suggests that the inactivation of NUPR1 impairs mitochondrial function and energy metabolism in cancer cells, increases ROS levels, and triggers a variety of cell death pathways, including apoptosis, autophagy, and necroptosis [77]. An increase in intracellular ROS acts as a primary signal that can react with intracellular iron to generate a more active ROS type-hydroxyl radical (HO•), thereby triggering ferroptosis [78,79]. Elsewhere, various ferroptosis inducers, such as erastin, RSL3 and sorafenib, were found to strongly activate NUPR1, supporting its protective role against ferroptosis [17,27].

### 4.1. NUPR1 Regulates Multiple Transcription Factors Involved in Ferroptosis

Evidence suggests the involvement of NUPR1 in ferroptosis through a variety of regulatory pathways. For example, RNF113A knockout dramatically reduced the NUPR1 overexpression induced by cisplatin and enhanced lipid peroxidation [80]. Interestingly, other studies have shown that DNA-damaged RNF113A-deficient cells undergo ferroptosis through the SAT1/ALOX15 signaling pathway [81,82]. Lipopolysaccharide (LPS), a macromolecule composed of lipids and polysaccharides, can cause strong leukocyte-infiltration, free radical production, and induce systemic inflammation [83,84]. In addition, it can disrupt mitochondrial DNA transcription, interfere with the oxidative phosphorylation (OXPHOS) process, reduce the levels of antioxidant enzymes such as glutathione S-transferase (GST) and superoxide dismutase (SOD), thereby inducing ferroptosis in bronchial epithelial cell lines [85,86]. Several studies have demonstrated the involvement of NUPR1 in this LPS-induced ferroptosis. For example, the pancreas showed instantly increased ROS levels and peroxidase enzyme myeloperoxidase (MPO) activity only within the first 6 h after LPS injection in wild-type mice [44]. In NUPR1 knockout mice, however, MPO activity and ROS levels induced by LPS were persistently increased and caused oxidative damage to the pancreas. Interestingly, the kinetics of MPO activity and ROS changes in the lung are the same as those in wild-type mice, indicating a tissue-specificity of NUPR1 in its regulatory function [44]. Furthermore, NUPR1 can transcriptionally regulate the key genes related to ferroptosis resistance, thereby conferring resistance to ferroptosis in cells [2,87,88]. Our latest research also found that by downregulating the expression of GSH-related genes, the NUPR1 inhibitor ZZW-115 affects the antioxidant system and promotes ferroptosis in cancer cells [89].

### 4.2. NUPR1 Controls Iron Metabolism

In addition to affecting the antioxidant system, NUPR1 also participates in iron metabolism in ferroptosis. It is involved in the regulation of the endogenous enzyme heme oxygenase-1 (HO-1) (Figure 2), which acts to neutralize ROS through its metabolite biliverdin, thus exerting an antioxidant effect [90]. Activation of the Nrf2/HO-1 signaling pathway protects kidney cancer cells from ER stress-related ferroptosis [91]. Similarly, overexpression of HO-1 attenuated the estrogen-mediated ferroptosis in kidney cells, while HO-1 knockout enhanced estrogen-induced ferroptosis in these cells [92]. Consistent with this, H_2_O_2_ dramatically promotes HO-1 reduction because of serum deficiency and induces oxidative damage, accompanied by NUPR1 activation [93]. Targeting NUPR1 prevents this process. Intriguingly, ferrous iron (Fe^2+^), a metabolite of HO-1, can react with intracellular ROS to generate HO•, which causes severe oxidative damage [94]. For example, BAY 11-7085 can induce ferroptosis in liver cancer cells by the overactivation of HO-1 [95]. Knockdown of HO-1 or the utilization of the HO-1-specific inhibitor zinc protoporphyrin IX (ZnPP) can significantly inhibit erastin-induced ferroptosis [96]. Consistent with these results, knockdown of NUPR1 activates HO-1 expression and increases cell viability in keratinocytes and mouse embryonic fibroblast (MEF) cells [97]. The antioxidant properties or pro-oxidant properties of HO-1 in different cells may depend on the balance between ROS and iron [98]. In this context, Liu and colleagues showed in cancer cells that NUPR1 can regulate iron metabolism through transcriptional activation of lipocalin 2 (LCN2) and protect the cells from ferroptosis [29]. Indeed, previous studies have also found that the silencing of NUPR1 down-regulates runt-related transcription factor 2 (RUNX2), a protein that is highly sensitive to iron overload, and ultimately causes premature senescence in breast cancer cells [57]. Therefore, NUPR1 can confer ferroptosis resistance to cancer cells both by enhancing the expression of antioxidant genes and iron metabolism.

### 4.3. NUPR1 Regulates Mitochondrial-Related Ferroptosis

The importance of NUPR1 in maintaining mitochondrial function is now well established, yet the exact role of mitochondria in ferroptosis remains elusive. While the classical ferroptosis inducers such as erastin and RSL3 do not increase mitochondrial ROS, a mitochondrial-targeting antioxidant mitoquinone (MitoQ) can protect against RSL3 toxicity [99]. However, mitochondrial membranes, being as abundant in polyunsaturated fatty acids (PUFAs) as the cell membrane, are also vulnerable to high ROS levels [100]. Indeed, changes in mitochondrial morphology such as condensed mitochondrial membrane densities, reducing mitochondrial volume, and damaged outer membrane are present in the cells treated with ferroptosis inducers [21]. Gao and colleagues found that although mitochondria play an indispensable role in cysteine deficiency-induced ferroptosis, they are dispensable for GPX4 inhibition-induced ferroptosis [101].

A series of recent studies underlined the importance of mitochondria in ferroptosis. The mitochondria-localized antioxidant enzyme dihydroorotate dehydrogenase (DHODH) can prevent oxidative damage to mitochondrial membrane lipids [102]. Its inhibition through brequinar sodium (BQR) induces mitochondrial-related ferroptosis in cancer cells with low expression of GPX4 and enhances the anticancer effect of ferroptosis inducers in cancer cells with a high expression of GPX4 [102]. TFAM is responsible for the replication and transcription of mitochondrial DNA and can reduce ROS production through Lon protease (LONP1) or the down-regulation of nuclear factor of activated T cells (NFAT) [103]. Zalcitabine, an anti-HIV medicine can induce ferroptosis by interfering with the LONP1/TFAM signaling pathway [104,105]. Similarly, our latest research also showed that the NUPR1 inhibitor ZZW-115 can inhibit TFAM expression, cause mitochondrial dysfunction, and produce more ROS production, thereby inducing ferroptosis in a variety of cancer cells such as pancreatic cancer cells and liver cancer cells. Interestingly, ZZW-115-induced ferroptosis can not only be prevented by ferroptosis inhibitors or antioxidants, but also by supplementation with TFAM (our unpublished results) (Figure 2). In summary, NUPR1 inactivation can induce ferroptosis via mitochondrial dysfunction, weakening of antioxidant capacity, and increasing the accumulation of endogenous iron content (Figure 2).

## 5. Conclusions

Regardless of histological type, all tumor cells have excessively high ROS levels and powerful antioxidant systems, which protect against oxidative damage [106,107]. Thus, compared with healthy cells, cancer cells are sensitive to the inhibition of antioxidants [108]. On one hand, as a stress protein, NUPR1 is strongly activated during oxidative stress. On the other hand, as an important transcription factor, NUPR1 can regulate the redox reaction for cancer initiation and development. Inactivation of NUPR1 thus changes multiple intracellular signals and affects cell function; it causes mitochondrial dysfunction, iron metabolic disorder, high ROS production and antioxidant defense impairment, all of which ultimately triggers ferroptosis in cancer cells. Therefore, NUPR1 is a crucial factor in the antioxidant system, and its targeting represents a promising strategy for cancer therapy.

Animal experiments have shown that genetic inactivation of NUPR1 suppresses different types of tumor growth, including pancreatic ductal adenocarcinoma (PDAC) [109], hepatocarcinoma (HCC) [17,110], lung adenocarcinoma [111,112], osteosarcoma [113], glioblastoma [114], cholangiocarcinoma [115], multiple myeloma [116,117,118] and ovarian carcinoma [119]. Among the inhibitors of NUPR1, ZZW-115 shows high potency and better affinity for NUPR1 compared to trifluoperazine (TFP) or other TFP-derived compounds, thus allowing stronger antitumoral activity [62]. The preclinical research demonstrated that ZZW-115 allowed dose-dependent tumor regression in HCC mouse models and different PDAC patient-derived xenograft (PDX) mouse models [62,76]. Importantly, no side effects were observed in these animal models upon ZZW-115 treatment [62]. In addition, NUPR1 can be activated by some anti-cancer agents, thereby conferring drug resistance to the targeted tumors [28]. Depletion of NUPR1 suppressed tumorigenesis and sensitized clear cell renal cell carcinoma to sorafenib treatment in vivo [120]. ZZW-115 treatment enhanced the in vivo anticancer activity of imidazole ketone erastin (IKE) [29] and induced a synergistic effect in combination with either 5-fluorouracil (5-FU) in pancreatic tumors or temozolomide (TMZ) in brain tumors [121]. Altogether, NUPR1 is a promising therapeutic target in cancer therapy. Further studies presenting the development of specific drugs targeting NUPR1 may bring enormous benefits to patients with cancer in the future.

## Figures and Tables

**Figure 1 cancers-13-03670-f001:**
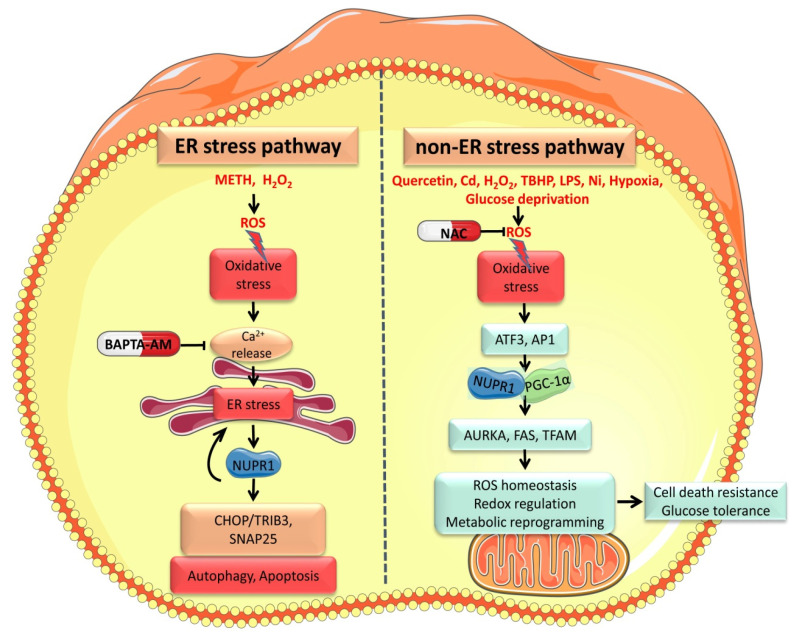
Oxidative stress activates NUPR1 by both ER stress (METH [11] and H_2_O_2_ [40]) and non-ER stress pathways (Quercetin [31], Cd [34], H_2_O_2_ [40,41], TBHP [42,43], LPS [44], Ni [45], hypoxia [46,47] and glucose deprivation [46,47,48].

**Figure 2 cancers-13-03670-f002:**
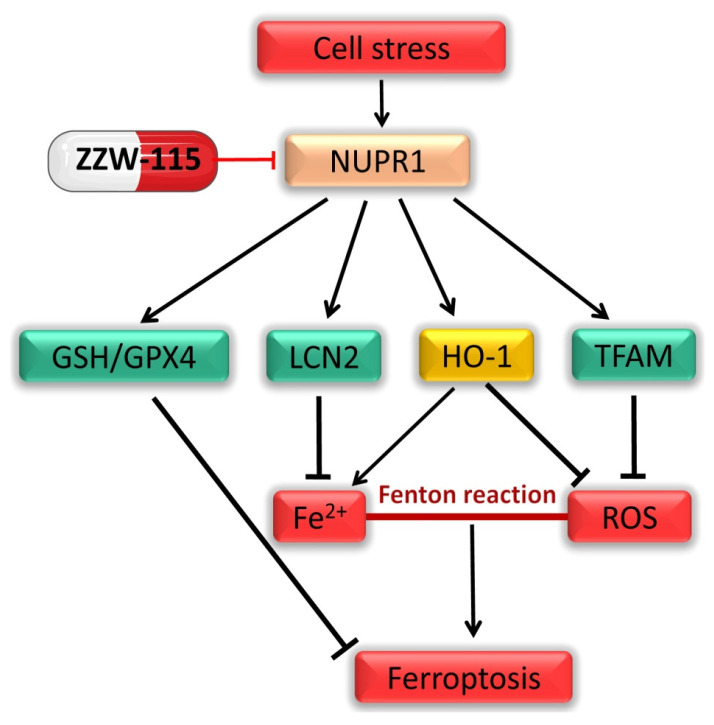
NUPR1 regulates ferroptosis via iron metabolism, ROS homeostasis and the GSH/GPX4 pathway.

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
