# Peer review of "NUPR1: A Critical Regulator of the Antioxidant System"

_cancers, 2021, doi:10.3390/cancers13153670_

Round 1
Reviewer 1 Report
In this article, the authors review the recent literature about the role of nuclear protein 1 (NUPR1) in the cellular defense against oxidative damage. Recent literature has extensively addressed the role of NUPR1 in cancer and as a potential drug target, and shed light on this protein. Therefore, in my opinion, a review of such literature would be of great interest for Cancers Journal readership. Also, the authors are good experts in the field and included in their article all the citations of the most recent findings about NUPR1.
Unfortunately, I found the manuscript difficult to read. There are multiple typos and changes to be made. Extensive language editing is necessary. Changes are necessary to improve its readability and to maximize its impact. Also, in some cases, the results of previous studies are described superficially.
The authors should extensively check their manuscript to improve the readability.
Please find some points below to be addressed.
Abstract
The abstract should be rephrased to better encourage readers to read the article. In the present version, it looks like a mere list of NUPR1 functions. I would include in the abstract a couple of sentences stating and discussing the pivotal role of NUPR1 in cancer and associated mechanisms and its potential role as a therapeutic target.
Introduction
The introduction is a little repetitive compared to the main body of the review. The authors should instead include in the introduction information about oxidant and anti-oxidant systems, ROS, ferroptosis, ER stress, and highlight the role of such stress responses in cancer, to bring all Cancers Journal readers “on the same page”. A conclusive sentence should be finally added stating the role of NUPR1 in these responses that will be detailed in the next paragraphs.
Please introduce in this section also the “NUPR1 downstream pathways” that are mentioned in line 77 for the first time and should have been introduced before. Describe the signaling pathways in which NUPR1 is involved.
Also, please pay attention to the phrase “NUPR1 is activated”. Unless specified, this statement is too general and readers cannot understand what it means. Does it refer to transcriptional activation? If so, please specify.
Figure 1
Please add a more extensive caption to the figure, commenting all the information contained and providing the corresponding references. For example, glucose deprivation is not mentioned in the text referring to the figure, nor in the caption.
Figure 2
Indicate the meaning of the color code, if any.
Main text
Line 52: “many” should refer to more than 4 studies. Please cite more studies, or change “many” to “various”
Line 61: “which that”, please correct
Line 66: this sentence is not clear, and, to my opinion, it is dispensable
Line 69: “which”, please correct
Line 92: “H2O2”, please correct to H2O2
Line 114: “;”, please correct
Line 151: “;”, please correct
Line 154: I would not say that mitochondria are mainly fueled by glucose, as amino acids and lipids also play a pivotal role in the regulation of energy metabolism in mitochondria.
Line 178: “it still has strong glucose tolerance”. Please correct
Lines 196-197: please uniform the verb tenses. Add -s for the third person if the verb is in present form
Line 207: “enhanced the lipid ROS production” please clarify. What do the authors mean with lipid ROS production? Lipid peroxidation?
Line 215: replace “more” with “several”
Line 216-219: please rephrase to make it easy to understand
Line 220: replace “persistent” with “persistently”
Line 226: “thereby affect” please correct
Line 235: H2O2, please correct to H2O2
Line 238-239: please correct
Line 242: replace Eastin with Erastin
Line 249: silence => silencing
General points
I think an additional paragraph could be highly beneficial to this article. The authors should dedicate a separate paragraph to the translational role of NUPR1. I would discuss preclinical data on NUPR1 as a therapeutic target, its role as a biomarker and in the resistance to therapy. This could be very helpful for Cancers Journal readers.
Please correct typos throughout the text.
Please uniform the verb tenses: choose the simple past of present form.
Please pay attention to abbreviations. For example, TFAM is specified only the second time it is used.
Author Response
Point‐by‐point responses to the reviewers
Reviewer 1
In this article, the authors review the recent literature about the role of nuclear protein 1 (NUPR1) in the cellular defense against oxidative damage. Recent literature has extensively addressed the role of NUPR1 in cancer and as a potential drug target, and shed light on this protein. Therefore, in my opinion, a review of such literature would be of great interest for Cancers Journal readership. Also, the authors are good experts in the field and included in their article all the citations of the most recent findings about NUPR1.
We thank the reviewer for these words.
Unfortunately, I found the manuscript difficult to read. There are multiple typos and changes to be made. Extensive language editing is necessary. Changes are necessary to improve its readability and to maximize its impact. Also, in some cases, the results of previous studies are described superficially.
The authors should extensively check their manuscript to improve the readability.
Please find some points below to be addressed.
Abstract
The abstract should be rephrased to better encourage readers to read the article. In the present version, it looks like a mere list of NUPR1 functions. I would include in the abstract a couple of sentences stating and discussing the pivotal role of NUPR1 in cancer and associated mechanisms and its potential role as a therapeutic target.
Thanks for this suggestion. We have included more relevant information regarding the pivotal role of NUPR1 in cancer, its associated mechanisms and its potential role as a therapeutic target.
Introduction
The introduction is a little repetitive compared to the main body of the review. The authors should instead include in the introduction information about oxidant and anti-oxidant systems, ROS, ferroptosis, ER stress, and highlight the role of such stress responses in cancer, to bring all Cancers Journal readers “on the same page”. A conclusive sentence should be finally added stating the role of NUPR1 in these responses that will be detailed in the next paragraphs.
We appreciate this suggestion. We have provided the information of these responses in cancer and added a conclusive sentence at the end of these sentences.
Please introduce in this section also the “NUPR1 downstream pathways” that are mentioned in line 77 for the first time and should have been introduced before. Describe the signaling pathways in which NUPR1 is involved.
This is a relevant suggestion. We have mentioned these pathways in the introduction section.
Also, please pay attention to the phrase “NUPR1 is activated”. Unless specified, this statement is too general and readers cannot understand what it means. Does it refer to transcriptional activation? If so, please specify.
We thank the reviewer for this suggestion. In fact it refers to the transcriptional activation. We have specified this outlined point in the revised version of the manuscript.
Figure 1
Please add a more extensive caption to the figure, commenting all the information contained and providing the corresponding references. For example, glucose deprivation is not mentioned in the text referring to the figure, nor in the caption.
We thank this reviewer for this comment. We are providing the corresponding references in the revised version of the text.
Figure 2
Indicate the meaning of the color code, if any.
We thank this reviewer for this comment. There is no indicated meaning of the color code.
Main text
Line 52: “many” should refer to more than 4 studies. Please cite more studies, or change “many” to “various”
We appreciate this suggestion. The text has been modified accordingly.
Line 61: “which that”, please correct
Thanks for this suggestion. The text has been corrected accordingly.
Line 66: this sentence is not clear, and, to my opinion, it is dispensable
We agree with this comment. We have modified the text according to the suggestion.
Line 69: “which”, please correct
We have corrected the text accordingly.
Line 92: “H2O2”, please correct to H2O2
We corrected it accordingly.
Line 114: “;”, please correct
Done.
Line 151: “;”, please correct
Done.
Line 154: I would not say that mitochondria are mainly fueled by glucose, as amino acids and lipids also play a pivotal role in the regulation of energy metabolism in mitochondria.
We agree with this suggestion. We have modified the text according to the suggestion.
Line 178: “it still has strong glucose tolerance”. Please correct
Thanks for your suggestion. We have modified the text accordingly.
Lines 196-197: please uniform the verb tenses. Add -s for the third person if the verb is in present form
We thank the reviewer for this suggestion. We have uniformed the verb tenses.
Line 207: “enhanced the lipid ROS production” please clarify. What do the authors mean with lipid ROS production? Lipid peroxidation?
We appreciate this suggestion. Text has been modified accordingly to this comment.
Line 215: replace “more” with “several”
Done.
Line 216-219: please rephrase to make it easy to understand
We thank the reviewer for this comment, and consequently we have rephrased this sentence.
Line 220: replace “persistent” with “persistently”
Done.
Line 226: “thereby affect” please correct
Done.
Line 235: H2O2, please correct to H2O2
Done.
Line 238-239: please correct
We have modified the text accordingly.
Line 242: replace Eastin with Erastin
Done.
Line 249: silence => silencing
Done.
General points
I think an additional paragraph could be highly beneficial to this article. The authors should dedicate a separate paragraph to the translational role of NUPR1. I would discuss preclinical data on NUPR1 as a therapeutic target, its role as a biomarker and in the resistance to therapy. This could be very helpful for Cancers Journal readers.
We thank this reviewer for her/his relevant suggestion. In the last section, we discuss the preclinical role of NUPR1 in cancer therapy.
Please correct typos throughout the text.
Done.
Please uniform the verb tenses: choose the simple past of present form.
We have checked and corrected the text according to this remark.
Please pay attention to abbreviations. For example, TFAM is specified only the second time it is used.
We thank the reviewer for this comment. We checked and modified abbreviations.

Reviewer 2 Report
Nicely written and timely review article on Nupr1 and its role as an antioxidant system.
Author Response
Point‐by‐point responses to the reviewers
Review 2
Nicely written and timely review article on Nupr1 and its role as an antioxidant system.
We thank this reviewer for these words.
Reviewer 3 Report
This interesting paper gathers the most recent research related to the activity NUPR1. Bibliography is relevant and recent, the paper is quite well written despite some typos and incorrectly phrased sentences, some of them being listed here bellow. I recommend that the authors read again the manuscript with great care to improve this or make use of an English language editing service. I think the manuscript should be published as it is when this editing problem is solved.
Detailed recommendations
Introduction: the nature of NUPR1 is not mentioned in the introduction. It appears very important to me that the authors state in this section of the manuscript that NUPR1 is a transcription factor (this is only being said in the conclusion of the paper) and maybe give a bit of insight into this function of the protein.
Line 43: typo “…was though to the only…” should read “was thought to be the only…”
Line 58: Cadmium symbol is Cd rather than CD
Line 61: typo “… CD led to lipid peroxidation and induced 60 NUPR1-dependent autophagy, which that also be inhibited by NAC”, should read “…which was also inhibited by NAC.”?
Line 69: typo “…from the ER, which inactivating Ca2+-dependent ER partners, and inducing the ER…”
Line 77-78: “…inhibits the rapamycin (mTOR) phosphorylation, and thereby promotes…” mTOR stands for “mechanistic target of rapamycin” and not “rapamycin” alone.
Line 80: “which can significantly inhibits elevated NUPR1 mRNA” should read “’which significantly inhibit” or “which can significantly inhibit”
Line 99: typo “… the elevanted levels …”
Line 130: typo “…NUPR1 129 transcriptional activated the presynaptic ROS sensor…”
Line 156 – 159: I think the following sentence need to be rephrased: “For example, in hypoxia or glu-156 cose starvation, NUPR1 knockout cells, γ-H2AX, a histone induces DNA damage through 157 NADPH oxidase 1 (Nox1) and ras-related C3 botulinum toxin substrate 1 (Rac1), signifi-158 cantly increased ROS production [45][46].”
Lines 156 – 160: More broadly regarding DNA damage and g-H2AX, I feel there is a little confusion between the DNA damage itself and the phosphorylation of the histone H2AX that is a consequence of DNA damage response (DDR) and is used as a proxy to assess DDR. This part may need to be more accurately formulated.
Lines 196 – 197: Please homogenize the verb conjugations:”… inactivation of NUPR1 impaired mitochondrial function and energy metabolism in cancer cells, increases ROS levels, and trigger a variety…”
Line 206 – 207: should read “…dramatically reduced the NUPR1 overexpression induced by cisplatin and enhanced…”
Lines 211 – 282: many typos/English language mistakes etc… please read again carefully to correct. (eg. “thereby affect” instead of “thereby affecting” l. 226 or “Eastin-induced ferroptosis” instead of “Erastin-induced ferroptosis” l.242)
Author Response
Point‐by‐point responses to the reviewers
Review 3
This interesting paper gathers the most recent research related to the activity NUPR1. Bibliography is relevant and recent, the paper is quite well written despite some typos and incorrectly phrased sentences, some of them being listed here bellow. I recommend that the authors read again the manuscript with great care to improve this or make use of an English language editing service. I think the manuscript should be published as it is when this editing problem is solved.
Thanks to the reviewer for these words. The text has been corrected by an English
Detailed recommendations
Introduction: the nature of NUPR1 is not mentioned in the introduction. It appears very important to me that the authors state in this section of the manuscript that NUPR1 is a transcription factor (this is only being said in the conclusion of the paper) and maybe give a bit of insight into this function of the protein.
Thanks for your suggestion. We have introduced the the nature of NUPR1 in this section.
Line 43: typo “…was though to the only…” should read “was thought to be the only…”
Thank to this reviewer for her/his suggestion. We have modified the text accordingly.
Line 58: Cadmium symbol is Cd rather than CD
Done.
Line 61: typo “… CD led to lipid peroxidation and induced 60 NUPR1-dependent autophagy, which that also be inhibited by NAC”, should read “…which was also inhibited by NAC.”?
Done.
Line 69: typo “…from the ER, which inactivating Ca2+-dependent ER partners, and inducing the ER…”
Done.
Line 77-78: “…inhibits the rapamycin (mTOR) phosphorylation, and thereby promotes…” mTOR stands for “mechanistic target of rapamycin” and not “rapamycin” alone.
Thanks to this reviewer for this remark. We corrected it in the revised version of the manuscript.
Line 80: “which can significantly inhibits elevated NUPR1 mRNA” should read “’which significantly inhibit” or “which can significantly inhibit”
We agree with this remark. We have modified the text according to this suggestion.
Line 99: typo “… the elevanted levels …”
Done
Line 130: typo “…NUPR1 129 transcriptional activated the presynaptic ROS sensor…”
The text has been modified accordingly.
Line 156 – 159: I think the following sentence need to be rephrased: “For example, in hypoxia or glu-156 cose starvation, NUPR1 knockout cells, γ-H2AX, a histone induces DNA damage through 157 NADPH oxidase 1 (Nox1) and ras-related C3 botulinum toxin substrate 1 (Rac1), signifi-158 cantly increased ROS production [45][46].”
This sentence has been rewritten.
Lines 156 – 160: More broadly regarding DNA damage and g-H2AX, I feel there is a little confusion between the DNA damage itself and the phosphorylation of the histone H2AX that is a consequence of DNA damage response (DDR) and is used as a proxy to assess DDR. This part may need to be more accurately formulated.
We agree with this reviewer. We have modified the text.
Lines 196 – 197: Please homogenize the verb conjugations:”… inactivation of NUPR1 impaired mitochondrial function and energy metabolism in cancer cells, increases ROS levels, and trigger a variety…”
We thank this reviewer for this remark. We have corrected the text according to this suggestion.
Line 206 – 207: should read “…dramatically reduced the NUPR1 overexpression induced by cisplatin and enhanced…”
Done
Lines 211 – 282: many typos/English language mistakes etc… please read again carefully to correct. (eg. “thereby affect” instead of “thereby affecting” l. 226 or “Eastin-induced ferroptosis” instead of “Erastin-induced ferroptosis” l.242)
We thank the reviewer for these remarks. We have checked and corrected the text according to her/his suggestions.